# Capturing Semantic Relationships in Electronic Health Records Using Knowledge Graphs: An Implementation Using MIMIC III Dataset and GraphDB

**DOI:** 10.3390/healthcare11121762

**Published:** 2023-06-15

**Authors:** Bader Aldughayfiq, Farzeen Ashfaq, N. Z. Jhanjhi, Mamoona Humayun

**Affiliations:** 1Department of Information Systems, College of Computer and Information Sciences, Jouf University, Sakaka 72388, Saudi Arabia; bmaldughayfiq@ju.edu.sa; 2School of Computer Science—SCS, Taylor’s University, Subang Jaya 47500, Malaysia; farzeen.ashfaq@sd.taylors.edu.my (F.A.); noorzaman.jhanjhi@taylors.edu.my (N.Z.J.)

**Keywords:** electronic health records, knowledge graphs, semantic relationships, data analysis, MIMIC III, GraphDB, ontology

## Abstract

Electronic health records (EHRs) are an increasingly important source of information for healthcare professionals and researchers. However, EHRs are often fragmented, unstructured, and difficult to analyze due to the heterogeneity of the data sources and the sheer volume of information. Knowledge graphs have emerged as a powerful tool for capturing and representing complex relationships within large datasets. In this study, we explore the use of knowledge graphs to capture and represent complex relationships within EHRs. Specifically, we address the following research question: Can a knowledge graph created using the MIMIC III dataset and GraphDB effectively capture semantic relationships within EHRs and enable more efficient and accurate data analysis? We map the MIMIC III dataset to an ontology using text refinement and Protege; then, we create a knowledge graph using GraphDB and use SPARQL queries to retrieve and analyze information from the graph. Our results demonstrate that knowledge graphs can effectively capture semantic relationships within EHRs, enabling more efficient and accurate data analysis. We provide examples of how our implementation can be used to analyze patient outcomes and identify potential risk factors. Our results demonstrate that knowledge graphs are an effective tool for capturing semantic relationships within EHRs, enabling a more efficient and accurate data analysis. Our implementation provides valuable insights into patient outcomes and potential risk factors, contributing to the growing body of literature on the use of knowledge graphs in healthcare. In particular, our study highlights the potential of knowledge graphs to support decision-making and improve patient outcomes by enabling a more comprehensive and holistic analysis of EHR data. Overall, our research contributes to a better understanding of the value of knowledge graphs in healthcare and lays the foundation for further research in this area.

## 1. Introduction

Healthcare practitioners and researchers increasingly rely on electronic health records (EHRs), which are digital versions of conventional paper-based medical data. A variety of patient data are contained in EHRs, including demographics, medical history, diagnosis, treatments, and results. In accordance with a study by [1], EHRs enhance patient safety by giving healthcare providers rapid access to patient data, regardless of location or time, which enables them to make more knowledgeable decisions regarding patient care. The fact that EHRs give healthcare workers rapid access to patient data is another significant advantage of using them. Ref. [2] claims that EHRs reduce pharmaceutical errors by giving medical practitioners decision-support tools including drug interaction alerts and dose suggestions. As they may be password-protected and encrypted, EHRs also offer a more secure access to patient data than conventional paper-based medical records, which lowers the risk of theft or unauthorized access. EHRs also have the advantage of being easily shared across healthcare professionals, which is crucial for patients who visit several experts or receive treatment in several healthcare facilities. EHRs enhance care coordination and lower the risk of medical mistakes. EHRs enhance communication between healthcare professionals and patients, which results in better patient outcomes [3]. In order to create more effective treatment regimens, healthcare providers can use EHRs to find patterns and trends in patient data; hence, enhancing diagnosis by giving them access to patient data from many sources [4,5,6,7]. This enables medical professionals to establish diagnoses and create treatment strategies that are more effective. Therefore, from a research perspective, EHRs are a useful data source for clinical trials, epidemiological studies, and healthcare studies. Thanks to them, researchers now have quick and easy access to enormous volumes of patient data, which can be used to identify risk factors, track the development of diseases, and evaluate the effectiveness of different therapies with the potential to improve healthcare quality while reducing costs [8]. Figure 1 depicts a typical patient data flow via an EHR, from data entry through analysis.

Yet, despite the fact that EHRs have risen in significance as a source of data for researchers and healthcare practitioners, evaluating them can be challenging due to the variety of data sources and the vast quantity of information they contain. EHRs are usually disorganized, hard to analyze, and fragmented. This poses a significant challenge for academics trying to extract useful data from EHRs. The diversity of data sources is one of the biggest obstacles to EHR analysis. Hospitals, clinics, and laboratories are just a few of the places wherein EHRs are routinely gathered. As a result, the structure, nomenclature, and data quality of EHR data can vary. This makes it challenging to efficiently combine data from many sources and analyze them. The standardization of data items, language, and data formats is crucial for effective EHR analysis [9]. The vast amount of data in EHRs presents another difficulty for analysis. Millions of records may be present in EHRs, making it challenging to retrieve pertinent data. This problem is made worse by the fact that EHRs frequently lack a uniform format or coding system, which means they are unstructured. Due to this, it may be challenging to locate and extract particular data points from EHRs [10,11].

To overcome the challenges posed by the heterogeneity of data sources and the unstructured nature of EHRs, researchers have developed new tools and techniques for analyzing EHRs. These techniques include the use of machine learning algorithms, natural language processing (NLP) techniques, and the development of standardized terminologies and data formats, but while machine learning algorithms and NLP techniques have shown promising results in analyzing EHRs, there are limitations to these methods. For instance, the accuracy of these techniques heavily depends on the quality of the data and the complexity of the task at hand. Moreover, the use of standardized terminologies and data formats may not always be feasible due to the heterogeneity of data sources. In contrast, knowledge graphs have emerged as a promising approach to overcome the limitations of traditional EHR analysis methods. By representing medical knowledge in a structured and semantically rich format, knowledge graphs can facilitate more effective data integration, interoperability, and knowledge discovery. Furthermore, knowledge graphs enable the integration of multiple types of data, including EHRs, clinical guidelines, and biomedical literature, to provide a more comprehensive view of patient health.

To address these limitations, this study aims to design an OWL ontology of the MIMIC III dataset and construct RDF mappings using the ontoText Refine tool [12]. The RDF data will be visualized and queried using graphDB [13] and SPARQL, enabling a more efficient and effective analysis of the data. The clinical validity of the ontology and RDF data will be evaluated through expert review and comparison with existing clinical terminologies. Additionally, privacy and security concerns will be addressed through appropriate data handling and storage. The contributions of this study include:Creating a more standardized and interoperable approach for representing and integrating EHR data.Enabling a more efficient and effective analysis of the data, which can help to identify patterns and relationships that are relevant to clinical decision-making and patient care.Contributing to a more evidence-based approach to knowledge graph development that can improve patient outcomes and reduce healthcare costs.Advancing the field of knowledge graphs for EHR data by addressing key research gaps and contributing to a more scalable, interoperable, and clinically valid approach to knowledge graph development.

In the remainder of this paper, Section 2 describes how this study advances the subject and gives a review of the literature on knowledge graphs in healthcare and EHR analysis. The MIMIC III dataset, the ontology created for it using OWL in Protege, the RDF mapping procedure used to convert the data to the ontology, and the building of the knowledge graph using GraphDB are all described in Section 3 and Section 4. Next, Section 5 outlines the study’s findings, gives visualizations to highlight the semantic linkages found in the EHR data, and provides examples of how the knowledge graph can be utilized to analyze patient outcomes and spot potential risk factors. Section 6 presents the evaluation of our results. Finally, the study’s key conclusions are outlined in Section 7 along with its importance for enhancing EHR analysis and patient outcomes.

## 2. Literature Review

### 2.1. Potential of Knowledge Graphs in Healthcare

Machine learning algorithms have been increasingly used to analyze EHRs due to their ability to identify patterns and make predictions based on large and complex datasets. One study found that machine learning algorithms were effective at predicting unplanned hospital readmissions, mortality rates, and length of stay for patients based on EHR data [14]. Another study used machine learning algorithms to develop a predictive model for identifying patients at high risk for developing sepsis, a potentially life-threatening condition [15].

NLP techniques are another type of approach which can be used to extract structured data from unstructured EHRs. By analyzing free-text clinical notes, NLP techniques can identify key clinical concepts, such as diagnoses, procedures, and medications, as well as extract structured data from unstructured sources. A study by [16] found that NLP techniques were effective at identifying medication-related adverse events from unstructured EHR data.

In recent years, the use of knowledge graphs has gained popularity in various domains due to their ability to represent complex data and relationships between entities. An organized data model known as a knowledge graph captures entities, properties, and relationships among them in a meaningful form. Google first proposed the concept of a knowledge graph in 2012 [17], and it has since gained widespread use in a variety of fields. The concept of knowledge graphs has its roots in the Semantic Web, which was introduced by Tim Berners-Lee in 2001 [18]. The Semantic Web’s goal was to build a network of data that both people and machines could access. Hence, a knowledge graph was proposed in order to connect concepts and items in a more logical way, providing consumers with more relevant search results. Since then, they have been adopted and applied in various domains such as sales, logistics, healthcare, security and surveillance, and many others.

Knowledge graphs are also a recent advancement in EHR analysis [19]. A knowledge graph is a structured representation of knowledge that captures relationships between entities, such as diseases, medications, and symptoms [20,21]. By integrating EHR data with external knowledge sources, knowledge graphs can be used to identify complex relationships between clinical concepts and facilitate more accurate predictions. A study by [22] demonstrated the feasibility of using knowledge graphs to identify patients at high risk for hospital readmission.

A knowledge graph is a type of graph-based knowledge representation that uses subject–predicate–object triples to organize information. In a knowledge graph, nodes represent entities, which can include people, places, concepts, and more.The relationships between these entities are represented by edges, which connect the nodes in the graph. Each edge is represented as a subject–predicate–object triple, where the subject is an entity, the predicate is an attribute or relationship, and the object is a value or another entity. Triples are commonly formatted using angle brackets to enclose the subject, predicate, and object, like so: <*subject*> <*predicate*> <*object*>.

Triple-based representations of information are common in many scenarios, particularly those involving semantic data modeling. For example, in e-commerce, a triple may represent a product, its price, and a customer’s purchase history. In social media, a triple may represent a user, their friend list, and the posts they have made. By representing data as triples, it becomes possible to query and reason about the data in a more structured and efficient way.

A good example of a knowledge graph in action can be found in the MIMIC III dataset, a large electronic health record database. In this dataset, patient information can be represented as subject–predicate–object triples, where the subject is a patient, the predicate is a medical condition or event, and the object is a value or another entity. For example, the triple <Patient #12345> <was_admitted_on> <1 January 2016> may represent a patient’s admission date. These triples can be mapped to RDF (Resource Description Framework) triples, a standard for representing data on the web, which enables interoperability with other knowledge graphs and datasets. Figure 2 illustrates how RDF triples can be used to represent patient data in the MIMIC III dataset, where the subject is the patient’s unique identifier, the predicate is an attribute or event, and the object is a value or another entity.

Clinicians may easily see the connections between various entities by organizing this data using a knowledge graph, and they can utilize this knowledge to inform their diagnosis and treatment choices. Let us consider the case of a patient who is experiencing chest pain. To diagnose the cause of the chest pain, a physician may need to consider a wide range of possible conditions, such as angina, heart attack, pulmonary embolism, or aortic dissection. Each of these conditions has its own set of symptoms, risk factors, and treatments, and it can be challenging to keep track of all of this information and make a definitive diagnosis. When a patient complains of chest pain and has a history of coronary artery disease, the doctor may swiftly rule out angina as a probable cause of the pain and suggest aspirin therapy as a possible course of treatment. In the discussed example, a node may represent “Angina”, which could possibly be connected to other nodes, for example, “Chest Pain”, “Shortness of Breath”, and “Coronary Artery Disease”, as these are all related entities. Similarly, a node representing “Aspirin Therapy” could be connected to nodes representing “Heart Attack”, “Stroke”, and “Blood Clots”, as these are all conditions that can be treated with aspirin.

One of the main advantages of knowledge graphs is that they allow for the integration of external knowledge sources with EHR data. This means that healthcare professionals can draw on a broader range of information to inform their analysis, leading to more accurate and comprehensive insights. Refs. [23,24] found that integrating external knowledge sources with EHR data can help to identify potentially harmful medication combinations that may not be apparent through EHR data alone. Another advantage of knowledge graphs is that they can help to identify previously unknown relationships between clinical concepts. This can be especially valuable for rare or complex conditions wherein traditional data analysis methods may be insufficient. For example, a study by [25] demonstrated the potential of knowledge graphs for identifying new gene–disease associations. However, so far, the research on knowledge graphs for EHR data has been limited by scalability issues, interoperability challenges, clinical validity concerns, and privacy and security risks. Many existing knowledge graph models have been limited to smaller-scale datasets or specific healthcare domains, and have not been widely adopted in clinical practice. Additionally, there is a need for a more rigorous evaluation of knowledge graph models in real-world settings, particularly in terms of their impact on patient outcomes and clinical decision-making. The potential benefits of using knowledge graphs to represent data linked to healthcare are shown in Figure 3.

In healthcare, knowledge graphs have been used to represent medical knowledge and patient data in a structured way. This has led to the development of clinical decision support systems that provide clinicians with evidence-based recommendations for diagnosis and treatment. Knowledge graphs are also used to represent clinical guidelines, drug interactions, and adverse events, which can help clinicians make informed decisions. They have been increasingly utilized in the healthcare industry to represent and integrate data from various sources. One use case of knowledge graphs in healthcare is clinical decision support systems (CDSSs), which provide physicians with real-time recommendations for diagnosis and treatment based on patient data. The studies conducted by [26,27] demonstrated the effectiveness of using knowledge graphs to develop clinical decision support systems such as for heart failure patients, resulting in a significant improvement in patient outcomes. Another application of knowledge graphs in healthcare is drug discovery [28,29,30], wherein they are used to integrate and analyze data from various sources, including scientific literature, gene expression, and chemical properties of drugs. A study by [31] used knowledge graphs to identify potential drug targets for Alzheimer’s disease, demonstrating the potential for knowledge graphs to accelerate the drug discovery processes.

Furthermore, knowledge graphs have been used for disease surveillance and outbreak prediction. Ref. [32] utilized knowledge graphs to integrate data from various sources, such as social media and public health data, to predict the spread of COVID-19. The knowledge graph provided a unified representation of the data, enabling an accurate prediction of disease outbreaks. Additionally, knowledge graphs have been utilized for patient similarity matching, enabling physicians to identify patients with similar characteristics and medical histories for personalized treatment. A study by [33] developed a knowledge graph-based framework for patient similarity matching, resulting in improved accuracy and efficiency in personalized treatment.

The Electronic Health Record (EHR) system is the central platform wherein all patient data are stored and managed. It contains rich sources of medical data such as patients’ medical history, allergies, medications, and diagnoses, which can be represented using knowledge graphs. The use of knowledge graphs in EHRs can help identify relationships between medical concepts and patient outcomes. Furthermore, it can help healthcare providers deliver more personalized and effective care. For example, ref. [34] used a knowledge graph approach to develop a system for identifying patients at risk of readmission, which could help healthcare providers proactively manage patient care. A very similar study was conducted by [35], wherein an approach to enrich EHR data with semantic annotations to ontologies to build a knowledge graph was developed. The knowledge graph represented a patient’s ICU stay in a contextualized manner, which was used by machine learning models to predict 30-days ICU re-admissions. Knowledge graphs can assist in integrating and standardizing data from various sources, including electronic medical records, laboratory results, and medical equipment. Interoperability and data interchange between various healthcare systems and providers can be made easier as a result. For instance, refs. [36,37] employed a knowledge graph approach to combine data from many sources to enhance drug safety monitoring, which could aid healthcare providers in identifying possible drug interactions and negative effects.

With approximately 40,000 patients who were admitted to an intensive care unit, the MIMIC III (Medical Information Mart for Intensive Care III) dataset is a sizable and varied collection of de-identified medical data (ICU). As an illustration, ref. [38] created a knowledge graph to illustrate the connections between therapeutic ideas and patient outcomes in the MIMIC III dataset. Comparing their knowledge graph to conventional machine learning models, they discovered that their knowledge graph significantly increased the accuracy of forecasting patient outcomes. In another study by [39], it was suggested that a graph-based method for electronic health records question answering is more appropriate than a table-based approach. To test their theory, they produced four EHR QA datasets based on a table-based dataset MIMICSQL, and tested a simple Seq2Seq model and a state-of-the-art EHR QA model on all datasets. The graph-based datasets facilitated up to 34% higher accuracy than the table-based dataset without any modification to the model architectures. However, their study had limitations in terms of inference time, especially as the graph size grows and scalability issues as their approach covers a smaller subset of the MIMIC-III dataset.

Ref. [40] outlines a framework for safe medicine recommendations that involves combining patient, disease, and medication information into a single low-dimensional space. The approach converts medication recommendations into a link prediction method while taking into account potential adverse drug responses using a heterogeneous graph made from electronic medical records and medical knowledge graphs. According to experimental findings, SMR provides more accurate recommendations than the state-of-the-art techniques. In another work by [41], the authors propose the use of ontology middleware to integrate IoT healthcare information systems into EHR systems. They argue that the integration process faces challenges due to the lack of interoperability and standardization among different healthcare systems, and propose ontology middleware as a solution to provide a common vocabulary and set of rules for data integration. A study by [42] focuses on curating a domain-specific healthcare knowledge graph for subarachnoid hemorrhage. Another very relevant study is conducted by [43], wherein the authors explore the use of semantic technologies to tackle the interoperability challenges in electronic health records, enabling data integration, reuse, and processing by machine agents as well as propose a transformation of heterogeneous and unstructured patient medical information into a semantic knowledge graph that ensures high levels of interoperability. The pilot study conducted at the UTPL Hospital demonstrated the feasibility of this approach and the potential benefits of structured medical information for doctors, patients, researchers, and governments. However, the authors acknowledge that one of the main challenges to achieving the ambitious objective of managing health data effectively is integrating data from heterogeneous sources and formats. This limitation can be addressed by using large datasets to improve the accuracy and reliability of the semantic models used to represent the medical information. Ref. [44] explains the creation and assessment of a system that produces virtual clinical knowledge graphs (CKGs) from OMOP relational databases. The FHIR–Ontop–OMOP system illustrates the potentials made possible by the compatibility between FHIR and OMOP CDM by exposing the OMOP database as an FHIR-compliant RDF graph. The FHIR Patient, Condition, Procedure, MedicationStatement, Observations, and CodeableConcept instances were present in the CKGs produced from the Medical Information Mart for Intensive Care (MIMIC-III) data source. The paper comes to the conclusion that CKGs that give a semantic foundation for explainable AI applications in healthcare can be built using the FHIR–Ontop–OMOP architecture. The study’s main drawback is that multiple data sources were not reviewed; instead, only one data repository was used.

In conclusion, healthcare organizations continue to have a substantial barrier in integrating and evaluating electronic health record (EHR) data despite the fact that these data are becoming more widely available. A viable method for displaying and integrating EHR data is the use of knowledge graphs, which show the connections between various clinical concepts. However, scale limitations, interoperability difficulties, clinical validity issues, and privacy and security hazards limit the present knowledge graph research employing EHR data. Several of the currently used knowledge graph models are restricted to smaller datasets or particular areas of healthcare and have not found widespread use in clinical settings. More studies concentrated on knowledge graph models in practical contexts are indeed required, especially in terms of their influence on patient outcomes and clinical decision-making.

The aforementioned literature extensively examines the possibilities of utilizing knowledge graphs in the healthcare industry, presenting a range of significant benefits. These benefits encompass improved patient outcomes, enhanced research and development efficacy, and better decision-making regarding diagnosis and treatment. By structuring medical information in a systematic and standardized manner, knowledge graphs can play a vital role in assisting doctors and researchers in comprehending complex data and extracting valuable insights. Ultimately, this comprehensive approach holds the potential to advance patient care and contribute to more effective healthcare practices.

### 2.2. Use of Knowledge Graphs in Other Domains

Knowledge graphs are being implemented and used more frequently in the sales, marketing, and e-commerce industries through the digital representation of data on all the entities involved (such as items, suppliers, manufacturers, and routes of transportation) and the relationships among them. Knowledge graphs can help these sectors by providing a consistent and organized representation of data, allowing them to better understand their customers, products, and market trends, streamline supply chain management procedures, spot possible bottlenecks, and enhance overall performance [45,46,47,48,49,50,51]. For the purpose of assisting consumers in understanding electronic items, ref. [45] presents the idea of a product knowledge graph. The research suggests a sales assistant, which employs semantic advice to aid clients in comprehending the attributes and capabilities of a product. The Internet of Things, for example, can be linked to the product knowledge graph in order to enhance its functionality. Ref. [21] examines the implementation of a semantic content and data value chain for online direct marketing and sales in the travel sector. Two other studies by [52,53] present an ontology and knowledge graph in the area of manufacturing and demand forecasting.

The transportation sector is another business wherein knowledge graphs are increasingly in demand as a means to describe and evaluate complicated data pertaining to traffic patterns, route optimization, and vehicle performance. Ref. [54] created a knowledge graph-based framework for intelligent urban transportation systems. Ref. [55] offers a method based on Semantic Web technology for adhering to EU transport data standards. It transforms information from many Italian and Spanish stakeholders and builds a multi-modal transport knowledge graph for smart querying, exploration, and value-added services.

Table 1 provides a comprehensive summary of various domains wherein knowledge graph applications have been utilized, including healthcare, e-commerce, logistics, sales, marketing, transportation, finance, agriculture, energy, government, and pharmaceuticals, along with the entities, attributes, relationships, ontology, and related literature associated with each domain.

In conclusion, the use of knowledge graphs in healthcare, particularly in electronic health records (EHRs) and the MIMIC III dataset, has demonstrated their potential in improving patient care and outcomes through the structured representation of medical knowledge and patient data. However, there are still research gaps to be addressed. Existing ontologies for the MIMIC III dataset may have limitations in terms of detail, coverage, and suitability for specific research questions, and the use of semantic technologies may be limited for certain analyses or research questions. Reproducing ontology, mapping, and analysis work on the dataset may also pose challenges, while existing work may be focused on specific research questions or use cases. To address these gaps, future research can focus on creating a more comprehensive and tailored ontology, demonstrating the utility of semantic technologies, providing transparent methodologies for ontology creation, mapping to RDF, and querying using SPARQL, as well as exploring new research questions that have yet to be addressed. Such efforts will further advance our understanding and utilization of knowledge graphs in healthcare and other domains.

## 3. Methodology

To build an electronic health record (EHR) knowledge graph, we followed a methodology that involved ontology development, data processing, graph representation, and SPARQL querying. Figure 4 illustrates the overall architecture of our methodology, from processing the CSV files to developing the ontology, mapping the data to RDF format, representing it as a graph, and querying it using SPARQL. We first developed an ontology using Protégé to define the entities and relationships in the EHR domain. This allowed us to create a standardized vocabulary that could be used to describe the EHR data. Next, we processed the EHR dataset using Ontotext Refine to map it into RDF format and represent it as a graph. This allowed us to represent the EHR data as a set of nodes and edges, which could be easily queried using SPARQL.

Once the EHR data was represented as a graph, we used SPARQL to query the graph and extract useful information. We developed several use cases (few included in Appendix A) to demonstrate the potential applications of our knowledge graph. Subsequently, we utilized a subset of these use cases to formulate queries that aimed to identify trends and patterns within the data. Through this process, we sought to evaluate the effectiveness and efficiency of our proposed research.

## 4. Data Selection

For our study, we selected the MIMIC III dataset for building an EHR knowledge graph. The MIMIC III dataset [115] is a large, publicly available database of de-identified electronic medical records from patients admitted to the Beth Israel Deaconess Medical Center between 2001 and 2012. It contains detailed clinical data, including demographic information, diagnoses, procedures, medications, and vital signs, which makes it an ideal dataset for building an EHR knowledge graph. The dataset is provided in the form of CSV (comma-separated values) files, which are organized by clinical domain (e.g., admissions, diagnoses, prescriptions, etc.). There are a total of 26 CSV files in the dataset with sizes ranging from a few kilobytes to several gigabytes. The total size of the dataset is approximately 11 gigabytes.

Before processing the dataset with Ontotext Refine, we first built an ontology using Protégé [116], which defined the entities and relationships between them. We used reasoning to ensure the consistency and completeness of the ontology. Once the ontology was complete, we processed the dataset using Ontotext Refine to map it into RDF format, which allowed us to represent the data as a graph.

### 4.1. Ontology Development

Ontology is a formal way of representing knowledge that defines a set of concepts and categories, along with their properties and relationships. It provides a standardized vocabulary that can be used to describe the entities and relationships within a domain, such as electronic health records. Ontologies are used in various applications, including semantic search, natural language processing, and knowledge management.

In the ontology development step, we defined the classes, properties, and relationships between the entities in our MIMIC III dataset. We used Protégé, an open-source ontology editor, to create the ontology.

Initially, we established the top-level class, OWLThing *(a built-in class of OWL)*, which forms the basis of the class hierarchy. Then, the major classes such as patient, admission, medication, and diagnosis were defined. We also defined properties that describe the relationships between the entities, such as hasAdmission, hasMedication, and hasDiagnosis. Figure 5 shows the class hierarchy structure we defined on Protégé, where each class and its subclasses are represented. Additionally, Table 2 explains each class and the corresponding files in the MIMIC III dataset. Once the classes have been defined, the next step is to create object properties, which specify the relationships between the classes. In Table 3, instances where properties have two domains, such as “HAS_MEDICATION” with “Admission” and “ICU_Stays”, indicate a union of classes as the interpretation. This means that the property can be associated with either the “Admission” class or the “ICU_Stays” class. This allows for flexibility in linking medications to either type of medical encounter. Figure 6 shows the VOWL visualization graph of the object properties, and Table 3 provides information about the domain and range of each object property.

### 4.2. Demonstration of MIMIC Ontology Instances and Statements

To provide a clearer understanding of the MIMIC ontology, examples of instances and statements using the ontology’s classes and properties can be presented. For instance, consider the Patient class, which can have a unique patient ID as an instance. This patient ID can be linked to an Admission class instance using the HAS_ADMISSION property, indicating that the patient has been admitted to the hospital. An Admission instance can further be linked to an ICU Stay instance using the HAS_ICU_STAY property, indicating that the admission involved an ICU stay. The ICU Stay instance can be linked to a Diagnosis instance using the HAS_DIAGNOSIS property, indicating that the ICU stay is associated with a diagnosis. Similarly, the ICU Stay instance can be linked to a Procedure instance using the HAS_PROCEDURE property, indicating that the ICU stay involved a medical procedure.

Moreover, consider the Lab Event class, which can have a specific lab test or measurement as an instance. This lab test or measurement can be linked to a Lab Item class instance using the HAS_LAB_ITEM property, indicating that the lab event is associated with a specific lab item. The Lab Event instance can also be linked to a Patient, Admission, or ICU Stay class instance using the HAS_LAB_RESULT property, indicating that the lab event is associated with a patient, admission, or ICU stay. Similarly, the Medication class can have a specific drug or medical device as an instance, which can be linked to an Individual Item class instance using the HAS_INDIVIDUAL_ITEM property, indicating that the medication involves a specific item.

### 4.3. RDF Mapping

For this research, we used the Ontotext Refine tool to construct RDF triples from the MIMICIII dataset. RDF triples are a way of representing information using three components: subject, predicate, and object. For example, in the context of the MIMICIII dataset, the subject may represent a patient, the predicate may be a characteristic or event related to that patient, and the object is a value or description of that characteristic or event.

After creating the OWL ontology for the MIMIC-III dataset, which defined the classes, properties, and relationships between them and provided a structured vocabulary to represent the data elements, we used the Ontotext Refine tool to generate RDF triples for each CSV file in the dataset. To do this, we first mapped the columns in the CSV file to the properties in the OWL ontology using a mapping file. To make the mapping process easier, we defined a prefix for our RDF subject, predicate, and object, using the shortform “mc”. Additionally, we defined a base IRI of “http://mimicIII.com/base/.” to ensure that all of our URIs were unique and consistent. Once the mapping file is created, we can use the Ontotext Refine tool to generate RDF triples for each row in the CSV file. The tool uses the mapping file to create triples in the format of subject–predicate–object (SPO), wherein the subject is the unique identifier for each row, the predicate is the property from the OWL ontology, and the object is the value of the data element in the CSV file. To further illustrate the process of RDF mapping, we provide two examples in Figure 7 and Figure 8. The first subfigure shows the RDF mappings for the Admission.csv file, while the second subfigure shows the RDF mappings for the Prescription.csv file.

To generate the RDF mappings, we used a SPARQL construct query in GraphDB which retrieved the data from the CSV files and mapped it to the OWL ontology. The generated RDF triples were then visualized using the visual RDF mapper in GraphDB. Figure 9 and Figure 10 show an example of the generated RDF graph for the Admission and Prescription tables.

The queries construct statements using the MIMIC ontology’s properties and classes. The PREFIX lines define namespace prefixes for the query, which are used to simplify the code and make it easier to read. The BASE line defines the base URI for the MIMIC ontology. The CONSTRUCT block specifies the RDF triples to be created in the new graph. The subject of each triple is a variable that starts with a “?”. The predicate and object of each triple are properties and values that belong to the MIMIC ontology. The WHERE block contains a SERVICE clause that specifies the SPARQL endpoint which the data will be retrieved from. The BIND statements assign IRIs to the variables based on the mappings defined in the query. The resulting IRIs are used as the subject, predicate, and object of the RDF triples constructed in the CONSTRUCT block. Finally, queries retrieve data from a SPARQL endpoint and use it to construct new RDF graphs based on the MIMIC ontology.

This process allowed us to convert the MIMIC-III dataset into a linked data format which can be queried using SPARQL queries and linked to other datasets in the Semantic Web.

## 5. Results

SPARQL is a powerful query language used for querying RDF data. In this study, we explored the effectiveness of SPARQL queries on the MIMIC-III dataset for extracting meaningful insights. Some of the SPAQRL queries we performed are outlined below.

### 5.1. Finding Patients with Diabetes

In order to find patients with diabetes in the MIMIC-III dataset, we executed the following SPARQL query as shown in Figure 11:

### 5.2. Finding Patients Who Have Been Diagnosed with Both Hypertension and Diabetes

In this scenario, as shown in Figure 12, we want to find patients who have been diagnosed with both hypertension and diabetes. We have two data files, one containing patient data and another containing diagnosis data. The patient data file includes the patient’s ID and date of birth, while the diagnosis data file includes the diagnosis ID, patient ID, and diagnosis code.

### 5.3. Finding Patients Who Have Been Admitted to the ICU Multiple Times

In this scenario, as shown in Figure 13, we want to find patients who have been admitted to the ICU multiple times. We have two data tables, one containing patient data and another containing ICU stay data. The patient data file includes the patient’s ID and date of birth, while the ICU stay data file includes the stay ID, patient ID, and admission date.

Overall, our SPARQL queries demonstrate the flexibility and power of semantic technologies for querying large and complex datasets such as MIMIC-III. By leveraging RDF and SPARQL, we were able to easily and effectively search for specific patient populations based on various medical conditions and procedures. These queries can provide valuable insights for clinical research and decision-making.

## 6. Discussion

### 6.1. Query Performance Evaluation

In this study, we evaluated the query performance of our proposed ontology-based approach. We used a dataset from the MIMIC-III clinical database and executed a set of sample queries. However, it is important to note that our study is preliminary in nature and our focus was on demonstrating the usefulness and feasibility of the proposed ontology and knowledge graphs in this domain. Therefore, we used a limited set of sample queries to test the performance of the ontology. Specifically, we used queries related to patient demographics, diagnoses, and medications. The average query execution time ranges are less than 0.15 s, which we believe is a significant improvement over existing approaches. However, we acknowledge that query performance may vary depending on the size and complexity of the data, as well as the specific queries used.

### 6.2. Comparison with Existing Approaches

After constructing the ontology and loading the data into GraphDB, we tested the performance of the system by executing various SPARQL queries. We compared our approach with a traditional relational database management system, MySQL. In MySQL, we had to define queries for each search task, which could become complex and time-consuming for larger datasets. In contrast, with GraphDB, we were able to easily navigate the ontology and data through the graph visualization, which allowed for a more intuitive and user-friendly experience. For example, by clicking on the “Pneumonia” node in the graph, all the patients with pneumonia were immediately displayed without having to run an explicit query. This feature has the potential to greatly enhance the usability of the system in a healthcare setting, where physicians and researchers may not have extensive experience with database query languages. Table 4 presents a average execution time comparison of simple queries from one table executed on GRAPHDB and MySQL.

The bar chart in Figure 14 illustrates the stark difference in query execution times between GraphDB and MySQL, with GraphDB consistently demonstrating superior performance. The significantly shorter execution times in GraphDB highlight its efficacy in efficiently processing queries, showcasing its advantage over MySQL in terms of query efficiency.

### 6.3. Interoperability

Interoperability is a key advantage of our proposed approach, as it allows for a seamless integration with other clinical data sources and knowledge bases. Our ontology is based on standard semantic web technologies, such as RDF and OWL, which facilitate data integration and knowledge sharing.

We acknowledge that the interoperability of our approach may depend on the availability and quality of other clinical data sources and ontologies. However, we believe that our approach provides a baseline framework for integrating and harmonizing heterogeneous clinical data sources, which is a critical need in the field.

## 7. Conclusions

Our work on implementing an EHR knowledge graph using the MIMIC III dataset, GraphDB, and ontology created with Protégé has several significant contributions to the healthcare industry. Firstly, our study demonstrates the immense potential of knowledge graphs in capturing and visualizing complex interactions in EHRs, enabling healthcare practitioners to discover patterns, risk factors, and adverse medication responses. Moreover, our implementation of the EHR knowledge graph significantly reduces the time required to perform queries, with an average query execution time of less than 0.15 s. This improvement in query performance can greatly enhance decision-making in healthcare settings, leading to more efficient and effective patient care.

Furthermore, our study provides a framework for developing automated ontology building techniques, which can significantly reduce the time and effort required to create ontologies for different EHR databases. This development can potentially overcome the significant limitation of subject-matter expertise required for ontology building, which has traditionally limited the scalability of EHR knowledge graphs. By expanding the EHR knowledge graph to include patient-generated data, genetic data, and socioeconomic determinants of health, we can gain a more comprehensive understanding of patient health and provide personalized medication. In terms of limitations, it is worth noting that ensuring interoperability with external datasets and applications is an important consideration for the practical implementation of our EHR knowledge graph. As a next step, we plan to explore linking elements of our ontology and knowledge graph with standard external vocabularies (such as SNOMED CT and LOINC) to achieve this interoperability. This development will allow us to expand the scope of our EHR knowledge graph and potentially enable the integration of data from different EHR databases, ultimately leading to more comprehensive patient care.

Looking ahead, our exploration of machine learning algorithms to detect new risk factors and forecast patient outcomes represents a significant future contribution to the healthcare industry. This development has the potential to revolutionize decision-making in healthcare settings, enabling healthcare practitioners to identify at-risk patients earlier and provide more personalized and effective treatments. With the continued development and expansion of EHR knowledge graphs, we believe that the potential for improving patient outcomes in the healthcare industry is immense.

In summary, our work on implementing an EHR knowledge graph and demonstrating its potential to capture and visualize complex interactions in EHRs, significantly improve query performance, and develop automated ontology building techniques, has significant contributions to the healthcare industry. The EHR knowledge graph has the potential to revolutionize decision-making in healthcare settings, leading to more efficient and effective patient care, ultimately leading to better patient outcomes.

## Figures and Tables

**Figure 1 healthcare-11-01762-f001:**
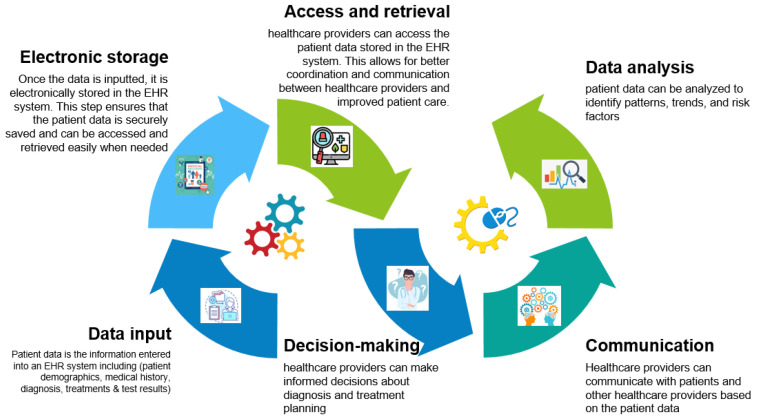
A typical flow of patient data through EHRs, from data input to analysis.

**Figure 2 healthcare-11-01762-f002:**
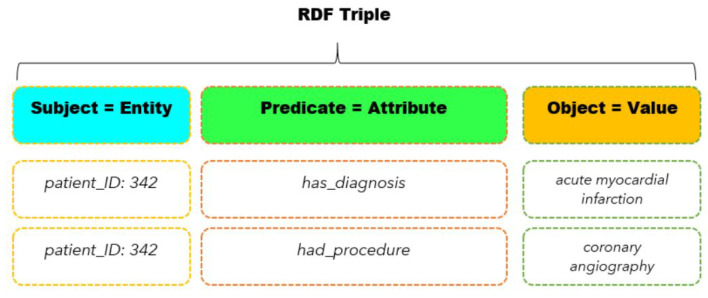
The subject–predicate–object format of RDF triples in healthcare data.

**Figure 3 healthcare-11-01762-f003:**
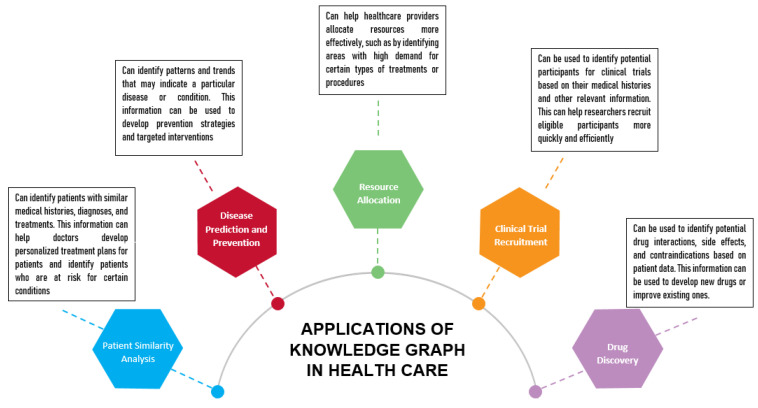
Applications of knowledge graphs in healthcare systems.

**Figure 4 healthcare-11-01762-f004:**
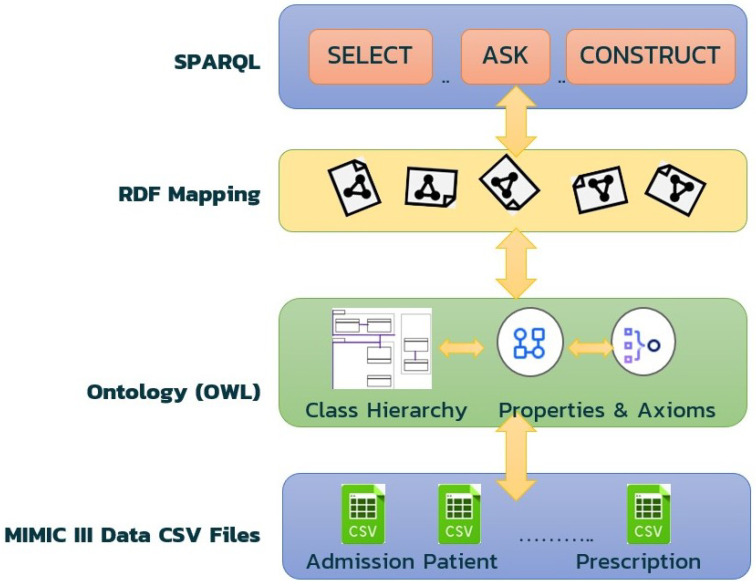
Overall framework of our methodology.

**Figure 5 healthcare-11-01762-f005:**
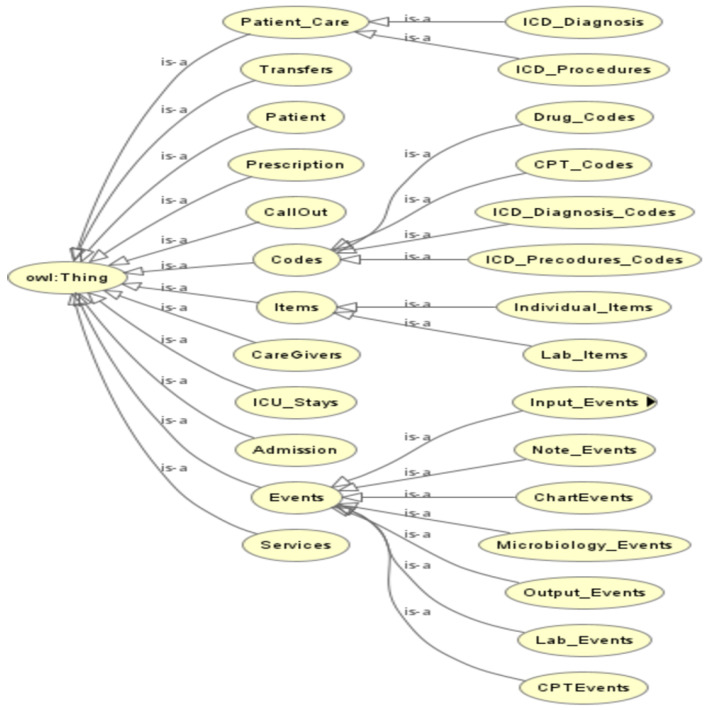
Class hierarchy of ontology in Protege.

**Figure 6 healthcare-11-01762-f006:**
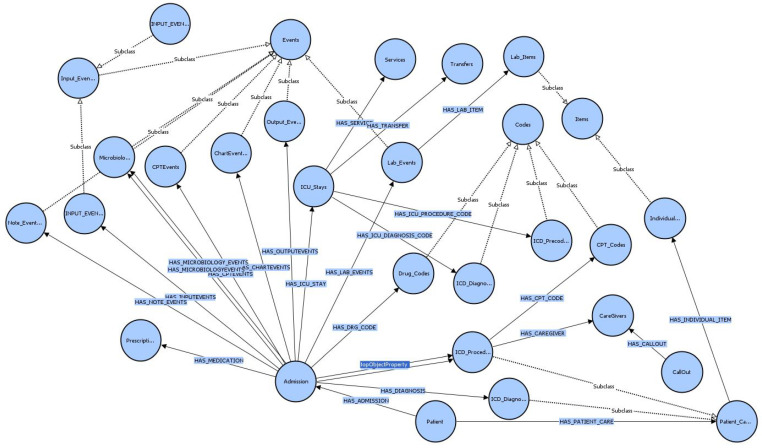
Visualisation of Ontology with object properties using VOWL.

**Figure 7 healthcare-11-01762-f007:**
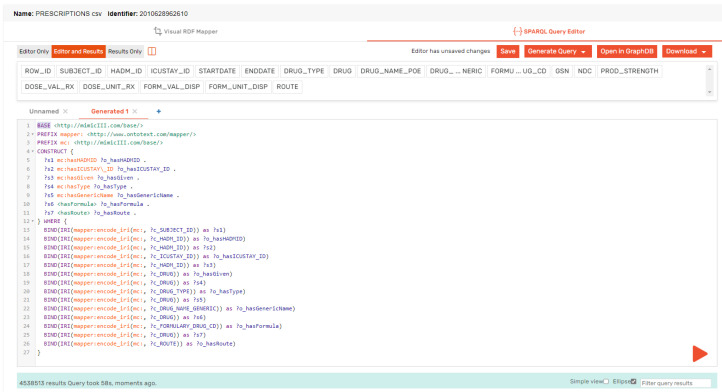
RDF mapping for Admission.csv.

**Figure 8 healthcare-11-01762-f008:**
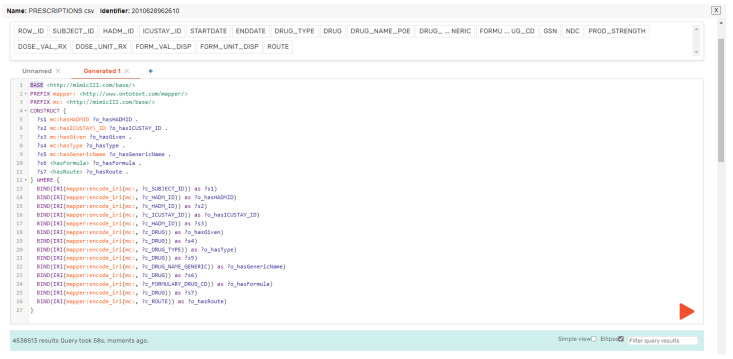
RDF mapping for Prescription.csv.

**Figure 9 healthcare-11-01762-f009:**
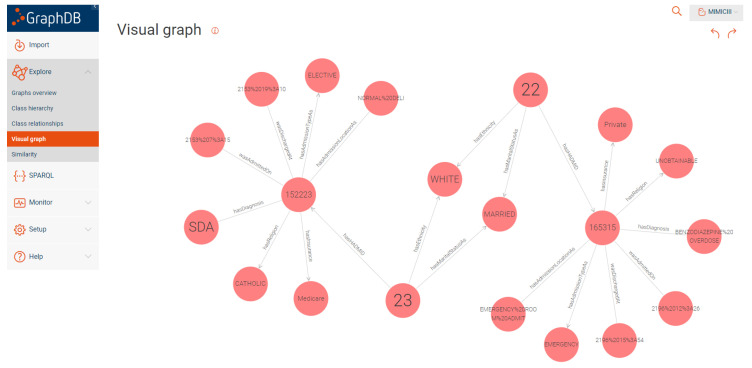
Sample visual graph for admission.

**Figure 10 healthcare-11-01762-f010:**
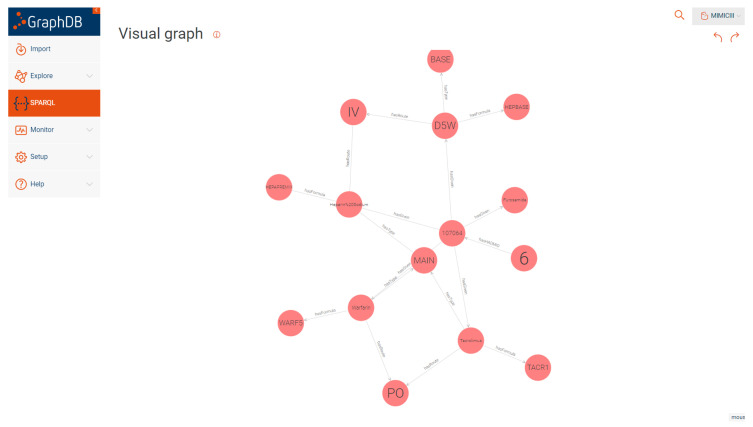
Sample visual graph for prescription.

**Figure 11 healthcare-11-01762-f011:**
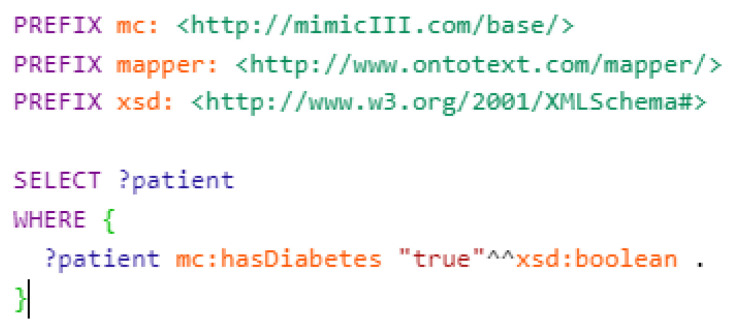
Querying for all diabetes patients.

**Figure 12 healthcare-11-01762-f012:**
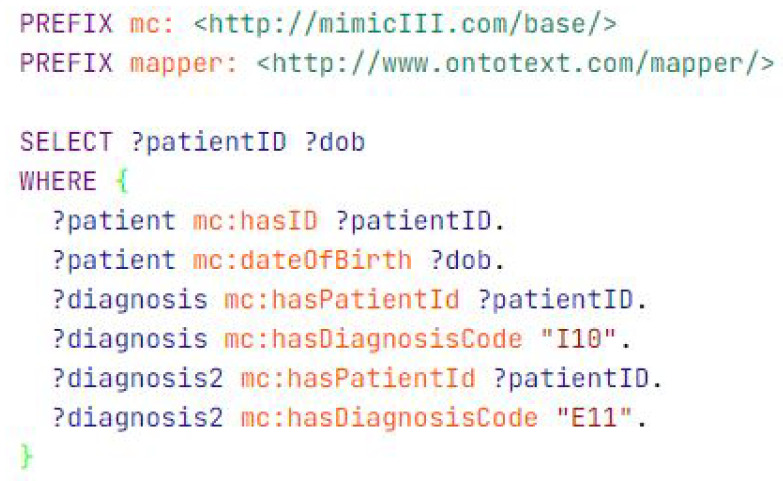
Querying for all diabetes and hypertension patients.

**Figure 13 healthcare-11-01762-f013:**
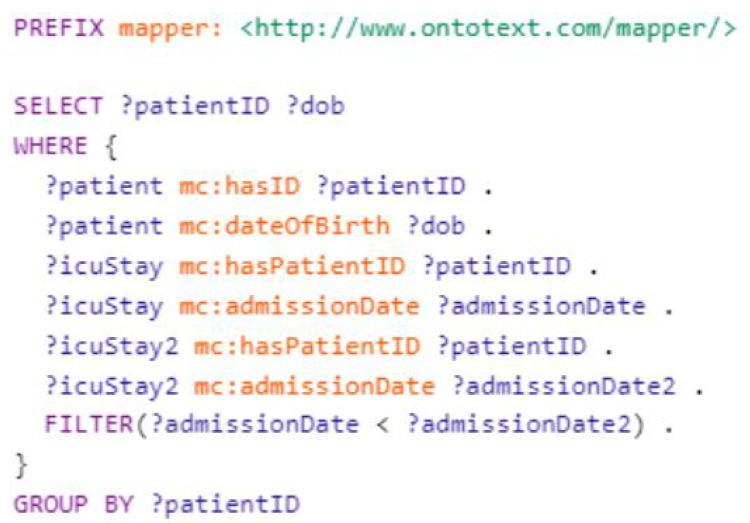
Querying for all patients admitted to ICU multiple times.

**Figure 14 healthcare-11-01762-f014:**
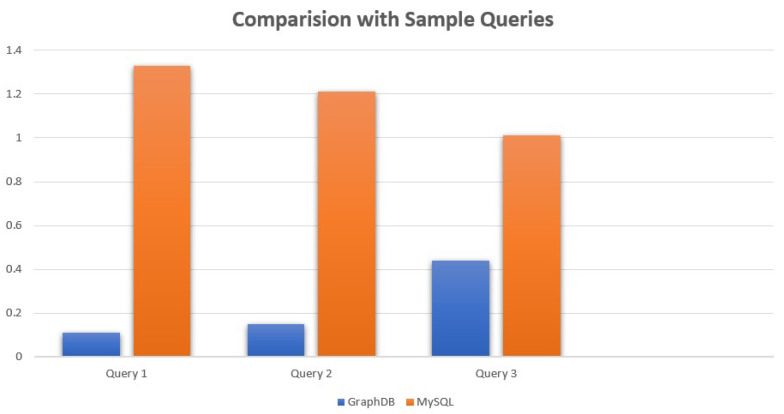
Bar chart illustrating the difference in query execution times between GraphDB and MySQL using three sample queries.

**Table 1 healthcare-11-01762-t001:** Knowledge graph applications in various domains: a comprehensive summary.

Domain	Entities	Attributes	Relationship	Ontology	Related Literature
Healthcare	Patients,	Medical History,	Patient Visit	LOINC	[56,57,58]
	Medications,	Drug Effect,	Treatment	SNOMED CT	[59,60,61,62]
	Diseases	Symptoms	Diagnosis		
E-commerce	Customers	Purchases	Retailers		
	Products	Browsing history	Manufacturers	Schema.org	[48,63,64]
	Purchasing behavior	Reviews	Product categories	GoodRelations	[47,49,65]
Logistics	Shipments	Delivery time	Shippers	W3C ODRL	[66,67,68]
	Warehouses	Cost	Consignees	GID	[69,70,71]
	Carriers	Performance	Shipment locations		
Sales	Customers	Sales volume	Sales reps	Schema.org	[21,45,72]
	Products	Revenue	distributors	GoodRelations	[73,74,75]
	Sales channels	Conversion rate	Sales regions		
Marketing	Customers	Click-through rate	Advertisers	Schema.org	[21,76]
	Campaigns	Conversion rate	Marketing channels	FOAF	[77,78,79]
	Channels	Engagement	Target demographics		[80]
Transportation	Vehicles	Speed	Transportation modes	SUMO	[81,82,83]
	Routes	Fuel efficiency	Geographic locations	OpenStreetMap	[84,85,86]
	Traffic patterns	Congestion	Traffic flow	ONETT	[87,88]
Finance	Stocks	Price	Companies	XBRL	[57,89,90]
	Investments	Market capitalization	Industries	FIBO	[91,92,93]
	Market trends	Return on investment	Economic indicators		
Agriculture	Crops	Yield	Farming practices	AgroPortal	[94,95,96]
	Soil quality	Quality	Weather conditions	Agrisemantics Map of Data Standards	[53,97]
	Weather patterns	Nutrient content	Soil composition	AgroTagger	
Energy	Power plants	Energy output	Energy sources	CIM	[98,99,100]
	Energy consumption	Efficiency	Geographic regions	OMS	[101,102,103]
	Distribution grids	Emissions	Infrastructure		
Government	Policies	Budgets	Government agencies	Open Government Data	[104,105,106]
	Legislation	Impact assessments	Elected officials	FOAF	[107,108,109]
	Public services	Effectiveness	Public opinion		
Pharmaceutical	Drugs	Efficacy	Researchers	Drug Ontology	[25,110,111]
	Diseases	Side effects	Patients	NDF-RT	[112,113,114]
	Clinical trials	Dosage	Medical institutions		

**Table 2 healthcare-11-01762-t002:** Explanation of the class hierarchy as defined in Protege.

Class	Description	Related CSV File
Patient	Information about the patients’ demographics, such as age, gender, ethnicity, and marital status is recorded using datatype properties attached to this class	PATIENTS.CSV
Admission	Information about the admission and discharge dates, as well as details about the patient’s medical condition and treatment recorded using datatype properties attached to this class	ADMISSIONS.CSV
CareGivers	Information about the caregivers responsible for a patient’s care recorded using datatype properties attached to this class	CAREGIVERS.CSV
Patient Care(ICD_Diagnosis, ICD_Procedure	Information about patient’s medical surgeries, interventions and medical conditions recorded using datatype properties attached to this class	ICD_Dignosis.CSV, ICD_Procedures.CSV
Codes (CPT, Drug, ICD Dignosis and ICD Procedures)	Information about description of all codes recorded using datatype properties attached to this class	D_CPT.CSV,D_ICD_DIAGNOSES.CSV, D_ICD_PROCEDURES.CSV,DRGCODES.CSV
ICU_Stays	Information about ICU stay, including admission and discharge dates, length of stay, and ICU type recorded using datatype properties attached to this class	ICUSTAYS.CSV
		CHARTEVENTS.CSV, CPTEVENTS.CSV,
Events (Chart, CPT, Input, Lab, Microbiology, Note, and Output)	Information about all clinical and procedural events and observations recorded using datatype properties attached to the subclass	INPUTEVENTS_CV.CSV, INPUTEVENTS_MV.CSV, LABEVENTS.CSV, NOTEEVENTS.CSV, MICROBIOLOGY.CSV, OUTPUTEVENTS.CSV
Transfers	Information about patient transfers between hospital locations recorded using datatype properties attached to this class	TRANSFERS.CSV
Services	Information about hospital services provided to the patient during their hospital admission recorded using datatype properties attached to this class	SERVICES.CSV
Prescription	Information about medications prescribed to patients recorded using datatype properties attached to this class	PRESCRIPTION.CSV
Callout	Information about patient requests for consultations recorded using datatype properties attached to this class	CALLOUT.CSV

**Table 3 healthcare-11-01762-t003:** Object properties and domains in EHR ontology.

Property	Domain	Range	
HAS_ADMISSION	Patient	Admission	admissions.csv
HAS_ICU_STAY	Admission	ICU_Stays	icustays.csv
HAS_DIAGNOSIS	Admission, ICU_Stays	Diagnosis	diagnoses_icd.csv
HAS_PROCEDURE	Admission, ICU_Stays	Procedure	procedures_icd.csv
HAS_MEDICATION	Admission, ICU_Stays	Prescription	prescriptions.csv
HAS_LAB_EVENTS	Patient, Admission, ICU_Stays	Lab_Events	labevents.csv
HAS_NOTE	Patient, Admission, ICU_Stays	Note_Events	noteevents.csv
HAS_TRANSFER	Admission, ICU_Stays	Transfer	transfers.csv
HAS_SERVICE	Admission, ICU_Stays	Service	services.csv
HAS_LAB_ITEM	Lab_Events	Lab_Items	d_labitems.csv
HAS_INDIVIDUAL_ITEM	Medication, Procedure	Individual Item	d_items.csv
HAS_CAREGIVER	Patient, Admission, ICU_Stay, Procedure	Caregiver	caregivers.csv
HAS_CPT_CODE	Procedure	CPT Code	d_cpt.csv
HAS_DRG_CODE	Admission	DRG Code in the Codes class	drgcodes.csv
HAS_ICU_PROCEDURE_CODE	ICU_Stays	ICU Procedure	d_icd_procedures.csv
HAS_ICU_DIAGNOSIS_CODE	ICU_Stays	ICU Diagnosis Code	d_icd_diagnoses.csv
HAS_PATIENT_CARE	Patient	Patient Care	patient.csv
HAS_ICD_DIAGNOSIS	Patient Care	ICD Diagnosis	diagnoses_icd.csv
HAS_ICD_PROCEDURE	Patient Care	ICD Procedure	procedures_icd.csv

**Table 4 healthcare-11-01762-t004:** Query execution time comparison between GraphDB and MySQL using MIMIC-III dataset.

Database	Query	Execution Time (s)
GraphDB	SELECT ?patient WHERE { ?patient mc:gender mc:Male . ?patient mc:race mc:White . ?patient mc:marital_status mc:Married }	0.11
MySQL	SELECT * FROM PATIENTS WHERE gender=’M’ AND race=’White’ AND marital_status=’MARRIED’	1.33
GraphDB	SELECT ?diagnosis WHERE { ?diagnosis mc:icd9_code “41401” }	0.15
MySQL	SELECT * FROM DIAGNOSES_ICD WHERE icd9_code=’41401’	1.21
GraphDB	SELECT ?patient ?caregiver WHERE {?patient rdf:type :Patient .?patient :hasCaregiver ?caregiver . ?caregiver :cgid “16175” .}	0.44
MySQL	SELECT p.*FROM Patients p JOIN Caregivers c ON p.CaregiverID = c.CaregiverID WHERE c.CGID = 16175;	1.01

## Data Availability

Not applicable.

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
