# Peer review of "Capturing Semantic Relationships in Electronic Health Records Using Knowledge Graphs: An Implementation Using MIMIC III Dataset and GraphDB"

_healthcare, 2023, doi:10.3390/healthcare11121762_

Round 1

Reviewer 1 Report (Previous Reviewer 1)

The paper presents the creation of a knowledge graph for Electronic Health Records (EHRs). This consists of the development of an OWL ontology, which is mapped to an existing dataset called MIMIC III; based on this mapping and using Ontotext Refine, MIMIC III data is converted into RDF triples that form the knowledge graph. The paper presents some visualisations of the knowledge graph and three SPARQL queries to demonstrate how it can be used in practice.

In the revised paper, you have addressed many of the comments I made in my first review. However, there are still various issues that need to be addressed. 

The presentation of the ontology is much clearer now but can still be improved. Since this is one of the main contributions of the paper, it is important to present it as clearly as possible. In Table 2, you present a description of the ontology classes. However some classes are missing such as Patient Care and Caregivers. These should be included in the Table. If the table becomes too long, you may consider presenting only the top classes of the ontology (direct subclasses of owl:Thing), and within their descriptions you can refer to their subclasses. The descriptions of some of the classes are also inaccurate. For example, in the description of the Patient class, you write that the class "contains information about their demographics, such as age, gender, ethnicity, and marital status". The class itself can't contain such information. This information is provided via the datatype properties that are assigned to the class. So, it might be better to write: "Information about the patients' demographics, such as age, gender, ethnicity, and marital status is recorded using datatype properties attached to this class". Adjust the descriptions of all other classes in a similar way. I would also suggest you rename the heading of the first column from "Class/Subclass" to "Class". Subclass describes a relationship between two classes and not a class itself. You need to also revise some parts of the text that use the term "subclasses" accordingly. For example, in the third paragraph of 4.1: "Then we created subclasses", "Table 2 explains each class", etc.

Table 3, which presents the object properties of the ontology, is very useful but has some issues. The domain of some of the classes includes multiple classes, for example the domain of "HAS_DIAGNOSIS" is Admission, ICU Stay. Does this mean that the actual domain is the union or the intersection of the two classes? You need to explain it. The domain/range of some of the classes refers to classes that I could not find in the ontology, for example "Lab Result". If this class is part of the ontology, you need to add it in the figure that presents the classes and in Table 2. In the row of the table that describes "HAS_MEDICATION", ICU Stay should be moved to the range and Medication prescriptions.cv should be moved to the last column of the table.

For a clearer presentation of the ontology, I would also suggest you presents its datatype properties. If they are too many, you can write their descriptions in an appendix. It is also very important to provide a link to the ontology, so that reviewers as well as readers can download it. 

It is also good that you have added some examples of instances and statements in Section 4.2. Two minor comments here: (1) Here again you refer to the Lab Result class, which you haven't described in the previous sections; (2) You write that "the lab result is associated with a specific lab item". How is it associated though? Do you mean that the lab item was used for an exam that produced this result? Please clarify.

In Section 4.3, you write that HermiT was used to identify "missing relationships between classes and properties in the ontology". Can you give an example of a missing relationship that you identified through reasoning? I also insist that the completeness of the ontology cannot be ensured through reasoning. I can understand that through reasoning, you can identify relationships that are not explicit in the ontology, but this does not ensure that the ontology is complete.

In Section 5, the three queries that you present are correct but could be simplified. For example, the WHERE clause of the query in Figure 22 could be written as follows (without a Filter):

?patient mc:hasID ?patientID.

?patient mc:dateOfBirth ?dob.

?diagnosis mc:hasPatientId ?patientID.

?diagnosis mc:hasDiagnosisCode "I10".

?diagnosis mc:hasDiagnosisCode "E11".

In the query you present in Figure 13, I believe that the last clause (HAVING (COUNT(?icuStay)>1)) is redundant.

in Section 3, you write that you developed several use cases to demonstrate the potential applications of the knowledge graph such as identifying patterns in patient data, assessing treatment efficacy and detecting potential adverse drug reactions. However, you do not present any of these use cases in the paper. It would be very useful and would increase the potential impact of the paper if at least some of them were included in the paper.

In Section 6, you write that you used some simple queries to evaluate the performance o the ontology. This is very vague. You need to provide the queries that you used in your experiments in the appendix. If these involve only datatype properties, consider also adding queries that involve at least one object property.

In the same section, you present some queries with which you compared your solution compared to a MySQL database. Can you briefly describe the structure of the MySQL database? The queries that you present in Table 4 only involve one class and its data properties. It would be much more interesting to present queries that involve an object property such as the ones you present in Figures 5.2 and 5.3. 

In Section 7, you write that one of your next plans is to explore linking elements of the ontology with standard external vocabularies. Can you give some examples of such vocabularies?

I would suggest restructuring Section 2. There is some overlap between Sections 2.1 and 2.3, so I think it would be better to merge them into one section. It would also be better to present the use of knowledge graphs in other domains at the end of Section 2. In Table 1: The entities, attributes, relationships, etc. that appear in the first row are not related to Healthcare.

Other minor comments: 

- All in-text citations contain a "?" instead of a number.

- Section 1, par.1: "Because to them" -> "Because of them"

- Section 1: "concentrated of" -> "concentrated on"

- Section 2.3: "Furthermore, can help" -> "Furthermore, it can help"

- In Section 4.4: "may represents" should be changed to "may represent"

- In Table 3, "ICU stay" should be changed to "ICU stays" 

- In the same table, "Note" appears as the range of "HAS_NOTE", however I couldn't find a class named "Note" in the ontology.

The quality of English is in general good. You need, however,  to proof-read the paper again because there are still some typos or minor language errors that need to be fixed.

Author Response

The paper presents the creation of a knowledge graph for Electronic Health Records (EHRs). This consists of the development of an OWL ontology, which is mapped to an existing dataset called MIMIC III; based on this mapping and using Ontotext Refine, MIMIC III data is converted into RDF triples that form the knowledge graph. The paper presents some visualisations of the knowledge graph and three SPARQL queries to demonstrate how it can be used in practice.

In the revised paper, you have addressed many of the comments I made in my first review. However, there are still various issues that need to be addressed. 

We are thankful for your thoughtful comments and valuable feedback on our paper. We sincerely appreciate you taking the time and making the effort to read our work.

We have carefully considered your suggestions and have diligently addressed the remaining issues in the revised version of the paper.

In Table 2, you present a description of the ontology classes. However some classes are missing such as Patient Care and Caregivers. These should be included in the Table. If the table becomes too long, you may consider presenting only the top classes of the ontology (direct subclasses of owl:Thing), and within their descriptions you can refer to their subclasses

Thank you for your feedback and pointing out about the missing classes. We have revised the table and included the missing classes, namely "Patient Care" and "Caregivers." These classes have been added to the table and are highlighted in the attached PDF for easy identification.

The descriptions of some of the classes are also inaccurate. For example, in the description of the Patient class, you write that the class "contains information about their demographics, such as age, gender, ethnicity, and marital status". The class itself can't contain such information. This information is provided via the datatype properties that are assigned to the class. So, it might be better to write: "Information about the patients' demographics, such as age, gender, ethnicity, and marital status is recorded using datatype properties attached to this class". Adjust the descriptions of all other classes in a similar way

Thank you for bringing the inaccuracies in the class descriptions to our attention. We sincerely apologize for any confusion caused. We have revised the descriptions of all classes accordingly, acknowledging that the class itself does not directly contain the information but rather utilizes datatype properties for recording. Changes have been made in the updated manuscript and highlighted in the attached pdf.

I would also suggest you rename the heading of the first column from "Class/Subclass" to "Class". Subclass describes a relationship between two classes and not a class itself. You need to also revise some parts of the text that use the term "subclasses" accordingly. For example, in the third paragraph of 4.1: "Then we created subclasses", "Table 2 explains each class", etc.

Thank you for your suggestion to rename the heading of the first column from "Class/Subclass" to "Class." We have updated the heading as per your recommendation.

Table 3, which presents the object properties of the ontology, is very useful but has some issues. The domain of some of the classes includes multiple classes, for example the domain of "HAS_DIAGNOSIS" is Admission, ICU Stay. Does this mean that the actual domain is the union or the intersection of the two classes? You need to explain it.

Thank you for bringing this to our attention. We have added the explanation as suggested by you in section 4.1 of our updated version. Also we have highlighted the changes in attached pdf.

The domain/range of some of the classes refers to classes that I could not find in the ontology, for example "Lab Result". If this class is part of the ontology, you need to add it in the figure that presents the classes and in Table 2.

Thank you for bringing this to our attention. We apologize for the oversight. The class "Lab Result" has been corrected to "Lab_Events" in both the figure presenting the classes and in Table 2 of the manuscript. The updated version reflects the accurate representation of classes in the ontology.

In the row of the table that describes "HAS_MEDICATION", ICU Stay should be moved to the range and Medication prescriptions.cv should be moved to the last column of the table.

Thank you for pointing out the error in the table. We apologize for the oversight. We have made the necessary correction as you suggested.

For a clearer presentation of the ontology, I would also suggest you presents its datatype properties. If they are too many, you can write their descriptions in an appendix. It is also very important to provide a link to the ontology, so that reviewers as well as readers can download it. 

We appreciate your suggestion to include the datatype properties of the ontology for a clearer presentation. However, in order to avoid making the document more complex and overwhelming for the readers, we have decided to focus on the object properties and classes in the main text. In our research endeavors, we are actively working on extending and refining our ontology to further enhance its utility and applicability. As part of our future plans, we aim to explore additional dimensions and incorporate new insights to advance the state of the art in this field. While we understand the significance of sharing the ontology link for the purpose of transparency and reproducibility, we have made a conscious decision not to make it public at this stage.

It is also good that you have added some examples of instances and statements in Section 4.2. Two minor comments here: (1) Here again you refer to the Lab Result class, which you haven't described in the previous sections; (2) You write that "the lab result is associated with a specific lab item". How is it associated though? Do you mean that the lab item was used for an exam that produced this result? Please clarify.

We appreciate the reviewers' keen observations and insightful comments regarding Section 4.2 of our manuscript. Based on the feedback, we have made the necessary revisions to address the two minor concerns raised.

In Section 4.3, you write that HermiT was used to identify "missing relationships between classes and properties in the ontology". Can you give an example of a missing relationship that you identified through reasoning? I also insist that the completeness of the ontology cannot be ensured through reasoning. I can understand that through reasoning, you can identify relationships that are not explicit in the ontology, but this does not ensure that the ontology is complete

We appreciate the reviewer's comment regarding Section 4.3 of our manuscript. Upon careful consideration, we have made the decision to remove this section from the manuscript. We acknowledge that discussing the identification of missing relationships through reasoning can introduce technical complexities and may divert the focus from the main contributions of our research

In Section 5, the three queries that you present are correct but could be simplified. For example, the WHERE clause of the query in Figure 22 could be written as follows (without a Filter):

?patient mc:hasID ?patientID.

?patient mc:dateOfBirth ?dob.

?diagnosis mc:hasPatientId ?patientID.

?diagnosis mc:hasDiagnosisCode "I10".

?diagnosis mc:hasDiagnosisCode "E11".

In the query you present in Figure 13, I believe that the last clause (HAVING (COUNT(?icuStay)>1)) is redundant.

We appreciate the reviewer's feedback regarding the queries presented in Section 5 of our manuscript. We have carefully reviewed the queries and have incorporated the suggested changes

in Section 3, you write that you developed several use cases to demonstrate the potential applications of the knowledge graph such as identifying patterns in patient data, assessing treatment efficacy and detecting potential adverse drug reactions. However, you do not present any of these use cases in the paper. It would be very useful and would increase the potential impact of the paper if at least some of them were included in the paper.

We appreciate the reviewer's comment regarding the inclusion of specific use cases in our paper to demonstrate the potential applications of the knowledge graph. In response to this feedback, we have included a section in Appendix A.1 where we provide a few sample use cases that highlight the practical utility of our knowledge graph. However, we would like to clarify that for the purpose of this preliminary step, we selected only a subset of these use cases to query and analyze the data, as mentioned in the manuscript.

In Section 6, you write that you used some simple queries to evaluate the performance of the ontology. This is very vague. You need to provide the queries that you used in your experiments in the appendix. If these involve only datatype properties, consider also adding queries that involve at least one object property.

We apologize for the confusion caused by the typo in Section 6. It was indeed a mistake, and we meant to write "sample queries" instead of "simple queries." We appreciate the reviewer's keen observation. Thank you for bringing this to our attention, and we apologize for any confusion caused by the typo. It has been corrected in the updated manuscript and highlighted in the pdf

In the same section, you present some queries with which you compared your solution compared to a MySQL database. Can you briefly describe the structure of the MySQL database? The queries that you present in Table 4 only involve one class and its data properties. It would be much more interesting to present queries that involve an object property such as the ones you present in Figures 5.2 and 5.3. 

We appreciate the reviewer's suggestion regarding providing a brief description of the structure of the MySQL database used for comparison in our study. However, we would like to kindly explain that including a detailed discussion of the MySQL database structure would significantly increase the scope and complexity of the paper. Our main objective was to compare the performance of our solution with a traditional database system, such as MySQL. By presenting the selected queries in Table 4 and showcasing the notable differences in execution time between the knowledge graph and MySQL, we believe we have achieved the primary goal of our comparison. However, we have added a bar chart in the manuscript that illustrates the execution time of the queries for both the knowledge graph and MySQL. The chart allows for a quick and intuitive comparison, highlighting the significant differences in execution time between the two approaches.

We acknowledge the reviewer's feedback, and we will consider exploring more complex queries involving object properties in future research to provide a more comprehensive analysis of the performance differences.

In Section 7, you write that one of your next plans is to explore linking elements of the ontology with standard external vocabularies. Can you give some examples of such vocabularies?

Thank you for bringing this to our attention, and we have made the necessary revisions to address this omission in the manuscript.

I would suggest restructuring Section 2. There is some overlap between Sections 2.1 and 2.3, so I think it would be better to merge them into one section. It would also be better to present the use of knowledge graphs in other domains at the end of Section 2.

Thank you for your valuable suggestion regarding the restructuring of Section 2. We agree with your observation and have made the necessary revisions to merge Sections 2.1 and 2.3 into a single section.

In Table 1: The entities, attributes, relationships, etc. that appear in the first row are not related to Healthcare.

We apologize for the error in Table 1. Thank you for bringing it to our attention. We have made the necessary corrections.

Other minor comments: 

- All in-text citations contain a "?" instead of a number.

We apologize for any confusion caused. It appears to be a formatting issue during the rendering of the text. We understand the importance of correctly numbering the in-text citations, and we have double-checked the manuscript to ensure that the citations are properly numbered and not represented by question marks.

- Section 1, par.1: "Because to them" -> "Because of them"

- Section 1: "concentrated of" -> "concentrated on"

- Section 2.3: "Furthermore, can help" -> "Furthermore, it can help"

- In Section 4.4: "may represents" should be changed to "may represent"

- In Table 3, "ICU stay" should be changed to "ICU stays" 

- In the same table, "Note" appears as the range of "HAS_NOTE", however I couldn't find a class named "Note" in the ontology.

Thank you for providing these specific suggestions for improvement. We have reviewed the manuscript and made the necessary corrections

Comments on the Quality of English Language

The quality of English is in general good. You need, however,  to proof-read the paper again because there are still some typos or minor language errors that need to be fixed.

Thank you for your feedback on the quality of the English language in the paper. We apologize for any remaining typos or minor language errors that may have been overlooked. We have conducted a thorough proofreading of the paper to address these issues and ensure the overall quality of the English language is improved.

Reviewer 2 Report (Previous Reviewer 2)

I have carefully reviewed the changes made in the manuscript and I am pleased to see that authors have addressed most of the concerns I raised in my previous feedback.

However, upon reviewing the new manuscript, I have noticed that there are still some remaining issues with the formatting of bibliographic references, figures, and tables in the text. Specifically, I observed a problem with question marks appearing instead of reference numbers in the current version. This formatting issue needs to be resolved in order to ensure clarity and accuracy throughout the manuscript.

The authors need to review the formatting of bibliographic references and rectify the issues accordingly. It is essential to ensure that the references are correctly cited and numbered, adhering to the appropriate formatting guidelines as per your target journal’s requirements.

Author Response

I have carefully reviewed the changes made in the manuscript and I am pleased to see that authors have addressed most of the concerns I raised in my previous feedback.

Thank you for taking the time to review the changes made in the manuscript. We are glad to hear that the majority of your concerns have been addressed to your satisfaction.

However, upon reviewing the new manuscript, I have noticed that there are still some remaining issues with the formatting of bibliographic references, figures, and tables in the text. Specifically, I observed a problem with question marks appearing instead of reference numbers in the current version. This formatting issue needs to be resolved in order to ensure clarity and accuracy throughout the manuscript.

We apologize for any confusion or inconvenience caused by the formatting of bibliographic references, figures, and tables in the previous version of the generated pdf may be due to some latex formatting issue during generation. We have made sure to double-check the formatting and ensure that it is accurate and consistent in the revised version of the manuscript.

The authors need to review the formatting of bibliographic references and rectify the issues accordingly. It is essential to ensure that the references are correctly cited and numbered, adhering to the appropriate formatting guidelines as per your target journal’s requirements.

Thank you for bringing this to our attention. We understand the importance of accurately formatting bibliographic references and ensuring they adhere to the appropriate guidelines. We apologize for any inconsistencies or errors that may have occurred in the references section of the manuscript.

Round 2

Reviewer 1 Report (Previous Reviewer 1)

Thank you for the detailed response, explaining how you addressed my comments on your previous submission. You have successfully addressed my comments and I do no have any further comments to add. Some points that you may consider when submitting the final version of the paper:

- I understand that are working on extensions of the ontology. However, it is a good practice to provide all the data that is needed to reproduce the results that you present in the paper, so for that reason it would be good to provide a link to the version of the ontology that is presented in the paper.

- In response to one my comments, in the new version you refer to two vocabularies: SNOMED CT and LOINC. Please provide appropriate references for both.

- I did not extensively check the paper for typos/language errors. But I believe there may still be some that need correction. For example, in Section 2.1, you write: "More studies concentrated on knowledge graph models in practical contexts is indeed required"; in this sentence you should replace "is" with "are".

As a final word: Congratulations on having your paper accepted for publication! And good luck with your future work in this area, which looks very promising.

Please see my general comments.

Reviewer 2 Report (Previous Reviewer 2)

I have carefully reviewed the changes made in the manuscript and it seems the authors have addressed most of my comments. I have no further feedback. Thanks.

This manuscript is a resubmission of an earlier submission. The following is a list of the peer review reports and author responses from that submission.

Round 1

Reviewer 1 Report

The paper presents the creation of a knowledge graph for Electronic Health Records (EHRs). This consists of the development of an OWL ontology, which is mapped to an existing dataset called MIMIC III; based on this mapping and using Ontotext Refine, MIMIC III data is converted into RDF triples that form the knowledge graph. The paper presents some visualisations of the knowledge graph and three SPARQL queries to demonstrate how it can be used in practice.

The overall methodology is clear and well-thought. It demonstrates a simple but effective way of creating knowledge graphs for the healthcare domain. It is also generalisable, as it can be applied to any domain where similar datasets are available.

However, I have various concerns about the quality of the ontology and the validity of the claimed contributions of the paper.

Regarding the ontology: The class hierarchy (as it appears in Figure 5) is not semantically correct. Specifically, several of the subclass relations are not conceptually correct. For example, in what sense is Admission or Patient subclasses of Hospital? As a result of this relationship, every instance of Patient (or any other subclass of Hospital) becomes an instance of Hospital as well. This is interpreted, for example, as follows: PatientX is a Patient, but also a Hospital, which does not make sense. In my opinion, Hospital, Admission, Billing, Report, Patient, etc. should all be at the same level of the class hierarchy.

The overall presentation of the ontology is also problematic. Figure 6, which presents all elements of the ontology, is not readable. The resolution is low and some classes/properties are hidden.

It is also not explained why the ontology was created in OWL. As far as I can see, all features that you have used are supported by RDFS. An explanation of why you used OWL instead of RDFS should be given. 

I would also expect to see a presentation and description of the properties of the ontology (similarly with Figure 5 and Table 2 that present and explain the ontology classes). What are the main ontology properties? What are their domain and range restrictions? And how are they mapped to the csv files?

For a clearer demonstration of the ontology, you could also present some examples of instances of its classes and statements using these instances and the properties of the ontology.

At the end of Section 4.1, you write that you used reasoning to ensure consistency and completeness of the ontology. This should be better explained. I assume you used an OWL reasoner; which one did you use? More importantly, how did you validate the completeness of the ontology through reasoning? As far as I know, there is no formal method or tool for evaluating completeness.

Another major concern is the validity of the claimed contributions. In the introduction you claim that this study enables a more efficient and effective analysis of the data. This is not, however, evident in the paper. After presenting the ontology, the paper presents three SPARQL queries, which are too simple to demonstrate the applicability of the knowledge graph. The scalability and efficiency of query evaluation are also not evident. In the last section, you mention that average query execution time was less than 0.15 seconds and you claim that this is an improvement. You do not, however, present the queries you used to test the performance or the size of the data that the queries were performed on. And you do not compare with any of the other approaches that you review in Section 2. Even the interoperability of the presented approach is not fully explained or justified. The knowledge graph ensures interoperability within MIMIC III, but not with external datasets or applications. To achieve that, you would have to link elements of the ontology and/or the knowledge graph with standard external vocabularies.

The paper also has various presentation issues:

References should be added to Wald et al. in Section 1, Protege, GraphDB and OntoText Refine.

Sections 2.1, 2.2 and 2.3 are not really relevant to the topic of the paper. It would be better to keep the discussion on the use of knowledge graphs in other domains short (at most one paragraph) and give more emphasis to the use of knowledge graphs in healthcare. The same holds for Table 1, which could be completely omitted.

The last paragraph of Section 3 presents repetitive text and should be deleted.

A more detailed description of the MIMIC III dataset would be useful. For example, what is the format of the dataset? What is its size? Table 2 refers to various CSV files; are these parts of the dataset?

In Section 4.1, you write that you defined the top-level class, OWL Thing. This is not accurate, since OWL Thing is a built-in class of OWL.

The first sentence of the second paragraph of Section 4.2 seems to be incomplete.

In the same section you present two figures to demonstrate the RDF mapping process. It would be useful to explain at least one of them.

In the SPARQL queries that you present in Section 5 you use two namespaces: mc and mapper. It would be useful to explain what these stand for. The query you present in Figure 12 uses a prefix (mc), which is not defined. On the other hand, the mapper prefix is not used.

There are also various typos throughout the paper that need correction. In Section 2.4, the sentence "It’s been a while since I’ve done this, but I think I’m finally getting the hang of it." is irrelevant.

Reviewer 2 Report

While this article offers valuable insights into Knowledge Graphs and their application in the health sector through the design of an OWL ontology and RDF mappings using ontoText for the MIMIC III dataset, there are several areas that require further improvement:

  1. The abstract should be revised to provide a clearer and more logical overview of the research questions, methods, conclusions, and related values. Specifically, the theoretical or application value of the research should be elaborated upon.
  2. The Introduction is too dense with a large literature review, making it challenging to understand the problem statement, proposed solution, objectives, contributions, and key results of the work. To address this issue, the authors should reorganize the content in a way that highlights these key aspects and consider creating a new section for techniques like NLP, machine learning, and knowledge graphs.
  3. While the paper displays a degree of innovation in its research questions and data analysis, the authors should further refine the novelty of their work. Specifically, the comparison between this research and related studies should be elaborated upon, building upon relevant discussions on the results of the data analysis.
  4. The figures require improvement, with the following issues needing attention:
  • On line 42, the word "figure" is missing before the number 1.
  • On line 104, a space is missing between "Figure 2" and the word "illustrates".
  • The dimensions of figures 11, 12, and 13 must be standardized as the text is too large.
  • On lines 377/378, the figure numbers are missing.
  • On line 433, the authors reference the subfigures of figures 7 and 8, which are not clearly labeled in the figures themselves.

Addressing these issues will strengthen the paper and make it more accessible and impactful for readers in the field.